# Molecular Detection, Seroprevalence and Biochemical Analysis of Lumpy Skin Disease Virus

**DOI:** 10.3390/v17030293

**Published:** 2025-02-20

**Authors:** Vandana Gupta, Annapureddy Pravalika, Megha Katare Pandey, Vineetha Mareddy, Anand Kumar Jain, Akansha Singh, Anju Nayak, Swati Tripathi, Shweta Rajoriya

**Affiliations:** 1Department of Veterinary Microbiology, Nanaji Deshmukh Veterinary Science University, Jabalpur 482001, India; pravalikaannapureddy@gmail.com (A.P.); swatitripathi134@gmail.com (S.T.); 2Department of Translational Medicine, All India Institute of Medical Sciences Bhopal, Bhopal 462026, India; meghapathology@gmail.com; 3Department of Pathobiology, University of Illinois Urbana-Champaign, Champaign, IL 61820, USA; reddyvineetha49@gmail.com; 4Department of Veterinary Physiology and Biochemistry, Nanaji Deshmukh Veterinary Science University, Jabalpur 482001, India; dranandjainsagar@gmail.com (A.K.J.); shwetarajoriya@gmail.com (S.R.); 5Department of Animal Genetics and Breeding, Nanaji Deshmukh Veterinary Science University, Jabalpur 482001, India; vetakki10@gmail.com

**Keywords:** LSD, seroprevalence, molecular detection, biochemical analysis

## Abstract

Lumpy skin disease (LSD) is a transboundary viral disease caused by lumpy skin disease virus (LSDV), belonging to the *Capripoxvirus* genus and *Poxviridae* family. This study reports on the molecular detection, seroprevalence and biochemical analysis of samples from cattle infected with LSDV in Madhya Pradesh (MP) and Telangana. A total of 189 samples (116 blood, 26 tissue, 47 nasal swabs) were collected from MP during 2022–2023. Molecular detection was performed using conventional PCR targeting the P32 and fusion genes, while seroprevalence was assessed using an indirect ELISA kit on 184 serum samples collected from MP and Telangana between 2022 and 2024. Tissue samples showed a higher positivity rate (69.23%) for the P32 gene, while nasal swabs had a 6.38% positivity rate. The fusion gene was detected in 77.77% of tissue and 66.66% of nasal swab samples. The seroprevalence study revealed that 19.56% of serum samples were positive, with a higher prevalence of 86.11% in MP. Biochemical analysis indicated elevated levels of SGPT, SGOT, BUN, creatinine, albumin, globulin and the A/G ratio in LSDV-infected cattle, though these differences were not statistically significant. The study emphasizes that blood samples are not ideal for LSDV detection and the timing of serum sample collection plays a critical role in seroprevalence studies.

## 1. Introduction

Lumpy skin disease is a vector-borne, re-emerging, transboundary disease that affects cattle and poses a serious threat to the livestock industry, resulting in significant financial loss on a global scale. It is listed as a notifiable disease by the World Organization for Animal Health [1]. First described in Zambia in 1929 [2], the disease was initially confined to southern and eastern Africa. However, it eventually spread across the entire continent, excluding Libya, Algeria, Morocco and Tunisia [3]. Later, it reached Europe and caused outbreaks in several Asian countries, including India. In India, the disease was first reported in Odisha in August 2019 [4] and subsequently spread to neighboring states such as Andhra Pradesh, Madhya Pradesh, Kerala and Assam during 2020–2021. By the end of 2022, it had also affected North-Western states like Gujarat, Rajasthan, Punjab, Haryana and Himachal Pradesh, as well as southern states such as Karnataka, Tamil Nadu, Kerala and Telangana.

Lumpy skin disease (LSD) is caused by lumpy skin disease virus (LSDV), which belongs to the genus *Capripoxvirus*, subfamily *Chordopoxvirinae* and family *Poxviridae* [5]. Capripoxviruses have a restricted host range and natural transmission between species other than sheep, goats and cattle does not easily occur [1]. LSDV is an enveloped virus with a distinctive dumbbell-shaped core and lateral bodies. Its genome consists of 151 kbp of double-stranded DNA, with inverted terminal repeats of approximately 2.4 kbp. The central portion of the DNA encodes 156 open reading frames (ORFs), which are annotated as putative genes and include 146 conserved genes necessary for replication, transcription and assembly [6]. Additionally, other members of the *Capripoxvirus* genus, such as goatpox virus (GTV) and sheeppox virus (SPV), share 97% genomic similarity with LSDV. This high similarity is responsible for the indistinguishable nature of these viruses in serological tests [6,7].

The virus is primarily transmitted by arthropod vectors, such as flies (*Stomoxys*), mosquitoes (*Aedes aegypti*) and certain ticks (*Rhipicephalus appendiculatus* and *Amblyomma hebraem*) [3]. Lumpy skin disease (LSD) is associated with high morbidity but low mortality [8]. Symptoms include fever, swollen lymph nodes, skin nodules and severe weight loss, leading to reduced milk production, infertility and significant pain. Over the course of 2 to 3 weeks, the nodules become necrotic and develop into “sit-fast” lesions, with necrotic material that eventually sloughs off, leaving cavities that can lead to bacterial infections [9]. LSD negatively impacts livestock economics by affecting meat and milk production, hide quality, draft power and reproductive efficiency. As a notifiable disease, it has serious implications for international trade.

Clinical lesions associated with LSD can be easily confused with those of other diseases due to similarities in clinical presentation. Differential diagnoses include bovine herpesvirus infections, actinomycosis, actinobacillosis, insect bites, foot-and-mouth disease (FMD), dermatophilosis, bovine viral diarrhea (BVD), rinderpest and nutritional deficiencies (such as selenium and copper deficiency), which can cause skin changes like roughness, hair loss and the formation of skin nodules. Therefore, accurate laboratory diagnosis is critical for the appropriate management and control of LSD outbreaks, as it helps to distinguish LSD from other diseases with similar clinical presentations and ensures that the correct interventions are implemented. Laboratory diagnosis of LSD can be performed using serological and molecular techniques, as well as virus isolation in cell cultures [10]. The present study aimed to detect LSDV from collected samples using conventional PCR, perform serological detection of LSDV antibodies by ELISA in cattle and determine changes in serum biochemical values in cattle infected with LSDV.

## 2. Materials and Methods

### 2.1. Study Area and Sample Collection

The proposed study was conducted in the Department of Veterinary Microbiology, College of Veterinary Science and Animal Husbandry, Nanaji Deshmukh Veterinary Science University, Jabalpur. A total of 189 samples (116 blood, 26 tissue and 47 nasal swab samples) were collected from animals exhibiting clinical signs such as fever, skin nodules and reduced milk production. These samples were collected from various districts of Madhya Pradesh (MP) during the period from 2022 to 2023 and stored in sterilized vials. The samples were subjected to molecular detection using conventional PCR, targeting the P32 and fusion genes. For the seroprevalence study, 184 serum samples were randomly collected from different districts of MP (116 samples) and Telangana (68 samples) between 2022 and 2024 in sterilized blood collection tubes. The serum was separated and stored at −20 °C for further analysis.

### 2.2. Genomic DNA Extraction

The LSDV genomic DNA was extracted from 189 collected samples from Madhya Pradesh (MP) using the QIAamp DNA Mini Kit (Qiagen, Hilden, Germany), following the manufacturer’s instructions.

### 2.3. Molecular Analysis

The extracted DNA was then subjected to PCR amplification using the GeneAmp PCR System 9700 (Applied Biosystems, Foster City, CA, USA). Initially, the samples were screened for the presence of the P32 gene and fusion genes of LSDV. The presence of LSDV was first confirmed using a specific PCR assay targeting the P32 gene, with the primer pairs forward primer 5′-TCCGAGCTCTTTCCTGATTTTTCTTACTAT-3′ and reverse primer 5′-TATGGTACCTAAATTATATACGTAAATAAC-3′, which amplifies a 192 bp fragment [11]. The PCR assay was performed using a 25 μL reaction volume, which included 12.5 μL of DreamTaq Green PCR Master Mix (2×) (Thermo Scientific, Leipzig, Germany), 1 μL of each primer (20 pmol/μL), 5.5 μL of deionized water, and 5 μL of DNA template. The PCR reaction was carried out in a Bio-Rad thermocycler (Bio-Rad, Hercules, CA, USA) under the following conditions: initial denaturation at 94 °C for 5 min, followed by 35 cycles of denaturation at 94 °C for 1 min, annealing at 50 °C for 30 s, extension at 72 °C for 1 min, and final elongation at 72 °C for 10 min. The PCR products were separated on a 1% agarose gel at 100 V for 25 min, and a 100 bp DNA ladder (Promega, Madison, WI, USA) was used as a size standard. The PCR products were visualized using a gel documentation system (Alpha Innotech, San Leandro, CA, USA).

The P32-positive samples were further screened for the presence of the fusion gene A27L using the following primers: forward primer 5′-AATGGA TCC ATG GAC AGA GCT TTA TCA ATC TTT C-3′ and reverse primer 5′-AAT GTC GAC TCA TAG TGT TGT ACT TCG GCC-3′ [12]. The PCR conditions for this assay were initial denaturation at 95 °C for 5 min, followed by 35 cycles of denaturation at 95 °C for 1 min, annealing at 60 °C for 40 s, extension at 72 °C for 1 min and final elongation at 72 °C for 7 min. The PCR products were visualized on a 1% agarose gel with a 100 bp DNA ladder.

### 2.4. Nucleotide Sequencing

The PCR products of the positive tissue samples for the envelope gene (P32) were sent for sequencing to Agri Genome Labs Private Limited, Kochi, Kerala, India.

### 2.5. Serological Analysis

All 184 serum samples were analyzed by indirect ELISA to detect the presence of LSDV antibodies using the commercially available AsurDx^TM^ Capripox Antibody Test Kit (Biostone, Dallas, TX, USA) following the manufacturer’s protocol. The results were interpreted by measuring the optical density (OD) values of the samples at 450 nm using an ELISA reader. The percentage positivity was calculated using the formula provided in the manufacturer’s guidelines.

### 2.6. Biochemical Analysis

Analysis of serum biochemical parameters was performed for 6 positive PCR samples using an autoanalyzer (Star 21 Plus). The following parameters were analyzed: creatinine; serum glutamic pyruvic transaminase (SGPT), also known as alanine aminotransferase (ALT); serum glutamic oxaloacetic transaminase (SGOT), also known as aspartate aminotransferase (AST); blood urea nitrogen (BUN); total protein (TP); albumin; globulin; and the albumin-to-globulin (A/G) ratio.

### 2.7. Statistical Analysis

For the seroprevalence study, the Chi-square test of significance was applied to evaluate the presence of lumpy skin disease virus (LSDV) antibodies in cattle serum samples. For the analysis of serum biochemical parameters, a *t*-test was used to assess differences in means between groups. Statistical analysis was performed using Statistical Package for Social Sciences (SPSS) version 25. A *p*-value of <0.05 was considered statistically significant for all tests.

## 3. Results

### 3.1. Molecular Analysis

When the samples were subjected to conventional PCR targeting the P32 gene, the expected product size was 192 bp (Figure 1), and for the fusion gene, the expected product size was 447 bp (Figure 2).

Molecular analysis of the P32 gene revealed a positivity rate of 69.23% (18/26) for tissue samples and 6.38% (3/47) for nasal swabs, with an overall positivity rate of 11.1% (21/189). None of the blood samples tested positive for LSD (Table 1).

For the fusion gene, 77.77% (14/18) of tissue samples and 66.66% (2/3) of nasal swab samples tested positive (Table 2).

The sequences of the P32 gene were submitted to the NCBI GenBank database under the accession numbers OQ183330.1, OQ150023.1, OR228441.1, OR228442.1, OR228443.1, OR228444.1, and OR228445.1.

### 3.2. Serological Analysis

The overall seroprevalence of lumpy skin disease virus (LSDV) was found to be 19.56% (36/184). The seroprevalence of LSD in Madhya Pradesh was 26.72% (31/116), while in Telangana, it was 7.35% (5/68). The samples collected from Madhya Pradesh were obtained during the peak outbreak period of LSD, and there was no prior history of immunization in the sampled population. In contrast, the samples from Telangana were collected during a period without an outbreak, and the animals had been immunized with the goatpox vaccine prior to sampling. This suggests that the timing of serum sample collection significantly influences seroprevalence results, particularly during active outbreaks.

The breed-wise prevalence of lumpy skin disease (LSD) in cattle was found to be 31.81% (7/22) in Holstein Friesian Cross, 25.86% (15/58) in Jersey Cross, 19.35% (12/62) in non-descript cattle, and 5.88% (1/17) in both Sahiwal and Gir cattle breeds. No positivity was observed in the Ongole Cross cattle (0/8). The breed-wise prevalence of LSD in cattle was found to be non-significant (*p* > 0.05) on statistical analysis (Table 3).

Age-wise analysis of seroprevalence for LSD indicated that the difference in prevalence was relatively higher in the adult group (21.33%, 16/75) compared to the calf group (19.23%, 10/52) and the young group (17.54%, 10/57), with no statistically significant variation (*p* > 0.05) (Table 3).

The sex-wise analysis of seroprevalence for LSD revealed that the prevalence in female cattle was slightly lower than in male cattle. Specifically, the seroprevalence in females was 19.25% (26/135), while in males, it was 20.4% (10/49). Statistical analysis revealed that the data were significant (*p* < 0.05) (Table 3).

The immunization status of the animals showed that the percentage positivity for lumpy skin disease virus (LSDV) antibodies in vaccinated individuals was zero, indicating no detectable antibodies against the field strain of LSDV. In contrast, a seroprevalence of 24.48% (36/147) was observed in non-immunized animals. Statistical analysis found the data to be significant (*p* < 0.05) (Table 3).

### 3.3. Biochemical Analysis

The serum biochemical values for LSDV PCR-positive samples are presented in Table 4.

A comparison of the biochemical parameters between the positive and negative groups was performed using a *t*-test, which showed no significant difference between the two groups. However, the values of some parameters, such as creatinine, SGPT, the A/G ratio, and total protein, were higher in the affected group.

## 4. Discussion

Molecular analysis of the P32 gene revealed a positivity rate of 69.23% (18/26) for tissue samples and 6.38% (3/47) for nasal swabs, with an overall positivity rate of 11.1% (21/189). The samples that were positive for the P32 gene were further screened for the presence of the fusion gene. For the fusion gene, 77.77% (14/18) of tissue samples and 66.66% (2/3) of nasal swab samples were found positive. These findings are consistent with those of Sanganagouda et al. [13], who reported a positivity rate of 95.04% for skin scabs and 66.66% for nasal swabs; Varadarajan et al. [14], who reported a rate of 87.5% for scab samples; Kumar et al. [15], reporting 80% for scab samples; Halmandge et al. [16], reporting 65.38% for skin biopsy samples; Geletu et al. [17], reporting 64.4% for skin scrap samples; and Sudhakar et al. [4], who observed a positivity rate of 50% for scab tissue and 20.45% for extended frozen semen.

None of the blood samples tested positive for lumpy skin disease virus (LSDV), which aligns with the findings of [18,19]. However, this contrasts with the findings of [20], who reported positive results of 2.5% in blood samples. No viruses were detected in the blood samples in our study, which is consistent with previous researchers’ work indicating that skin tissues, nasal swabs, and saliva are more effective for LSDV genome detection than blood samples [21]. This is likely due to the short viraemia period of LSDV, which typically lasts between 6 and 15 days post-infection [1], with the virus being epitheliotropic.

In this study, the overall seroprevalence was reported as 19.56% (36/184). This is higher than the seroprevalence reported by Abera et al. [22] (6.43%), Ochwo et al. [23] (8.7%), and Hailu et al. [24] (7.4%). However, the seroprevalence in this study is lower than those of [25,26], where the authors reported seroprevalence rates of 27% and 25.4%, respectively. These variations in seroprevalence levels may be attributed to several factors, including differences in population densities and the efficiency of arthropod vectors, dissimilar environmental conditions, cattle population characteristics, the timing of sample collection, and the testing methodologies employed in each study.

In this study, exotic breeds exhibited a higher prevalence of lumpy skin disease (LSD) compared to indigenous breeds. This finding is consistent with the study in [22], where higher seroprevalence was observed in crossbred cattle compared to local zebu cattle. However, in our study, breed-wise analysis was found to be statistically non-significant, which aligns with the findings of [23], who reported no association between cattle breed and LSD seropositivity. This contrasts with the study in [22], where a significant association was found between breed and seropositivity. The increased susceptibility in crossbred cattle might be due to several factors, including genetic differences that influence immune response, as well as management practices, since crossbreds are often raised in intensive farming systems. The higher stress levels associated with intensive farming can further compromise their immune response. Additionally, crossbred cattle may be more reliant on commercial feeds, which might not provide optimal nutrition for disease resistance.

Analysis of the association between age and seropositivity for lumpy skin disease (LSD) revealed no statistically significant variation among the three age groups, which is consistent with the findings of [22,27]. However, seroprevalence was higher in adult cattle compared to calves and young cattle, aligning with the results of [22,23,26]. The low seroprevalence in calves may indicate the presence of passive maternal immunity and limited exposure to the virus. Additionally, calves may be less susceptible to fly bites [28], and those kept at homesteads, where insect vector activity is lower, showed the lowest prevalence. Higher seroprevalence in adult cattle could also be due to a longer time period for possible exposure than younger animals, assuming outbreaks had already occurred several years previously.

The current study revealed a significant association (*p* < 0.05) between sex and seropositivity to lumpy skin disease virus (LSDV), with male animals showing a slightly higher seropositivity rate of 20.4% (10/49) compared to females at 19.25% (26/135). These findings contrast with those of [22,23,27], who reported higher seroprevalence rates in female cattle. This discrepancy may be attributed to differences in sample size and management practices. Despite the observed variation, it is important to note that sex may not play a critical role in the overall susceptibility to LSDV infection, suggesting that other factors, such as age and management practices, could have a more significant impact on determining seropositivity.

Limited information is available on the serum biochemical profiles of cattle infected with lumpy skin disease virus (LSDV). This study investigates serum biochemical changes in cattle infected with lumpy skin disease virus (LSDV), revealing elevated levels for SGPT, SGOT, BUN, BIT, creatinine, albumin, globulin and the A/G ratio, although most changes are statistically non-significant (Table 5). The increases in SGPT (ALT) and SGOT (AST) levels may reflect liver and cardiac damage, as these enzymes are found in hepatocytes, skeletal muscle, and cardiac muscle. Elevated SGOT levels, despite being non-significant in this study, are consistent with previous reports suggesting hepatocyte damage and muscle tissue breakdown due to LSDV or secondary infections [29]. Increased BUN levels likely result from dehydration and tissue damage, while slight increases in creatinine point to potential kidney involvement.

Total serum protein concentrations showed a non-significant increase, aligning with some studies but differing from others that reported significant changes. Variations across studies could be attributed to factors such as sample timing, geographic location, LSDV strain, and animal health status. The discrepancies in serum protein levels underscore the complexity of interpreting biochemical changes in LSDV-infected cattle and highlight the need for further research to understand the full impact of the virus on cattle physiology.

## 5. Conclusions

The livestock sector in India plays a crucial role in the country’s economy, with a cattle population exceeding 300 million. In 2022, a major outbreak of lumpy skin disease (LSD) led to an estimated economic loss of USD 2217.26 million, primarily due to factors such as livestock mortality, reduced milk production, abortion, infertility, and damaged hides. This study offers valuable insights into LSD by emphasizing the importance of molecular detection and seroprevalence studies, which are essential for selecting appropriate samples for accurate diagnosis. Additionally, this study highlights the breed predisposition to LSD infection, with indigenous cattle breeds showing greater resistance to the disease compared to exotic breeds. This research further underscores the critical role of vaccination in preventing LSD outbreaks. However, it is vital to carefully consider the type of vaccine used, as different vaccine strains may generate varying levels of immunity. Assessing how each vaccine strain contributes to the overall immune response is essential for selecting the most effective vaccine for disease control. Further investigation into the biochemical profiling of LSD, particularly with a larger sample size, is necessary to gain a deeper understanding of the disease’s pathogenesis. Ongoing monitoring of LSD strains in the field, along with tracking genetic variations from vaccine strains, is crucial for refining vaccination strategies. Generating sequence data from prevalent field strains will be instrumental in identifying potential differences and enhancing control measures. In conclusion, this study provides critical insights for managing and controlling LSD outbreaks in India. It highlights the need for continued research into vaccine efficacy, breed resistance, and strain variation to improve disease prevention and minimize the impact of LSD on the livestock sector.

## Figures and Tables

**Figure 1 viruses-17-00293-f001:**
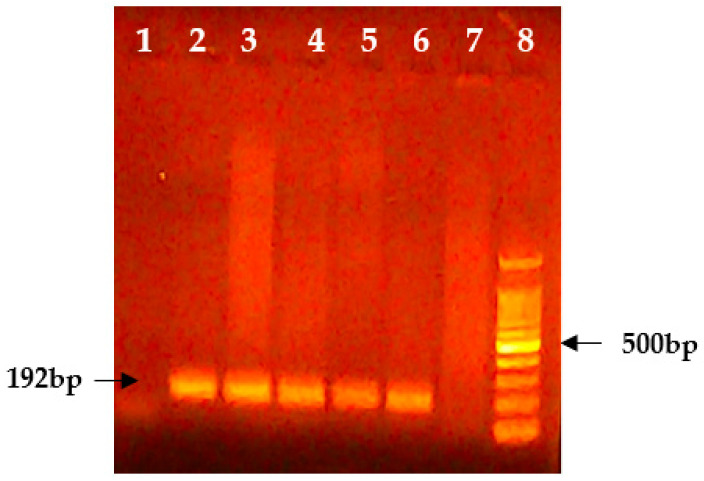
PCR gel picture of the P32 gene (192 bp). Lane 1: Negative control; Lanes 2, 3, 4, 5, and 6: Positive samples; Lane 7: Negative sample; Lane 8: DNA ladder (100 bp).

**Figure 2 viruses-17-00293-f002:**
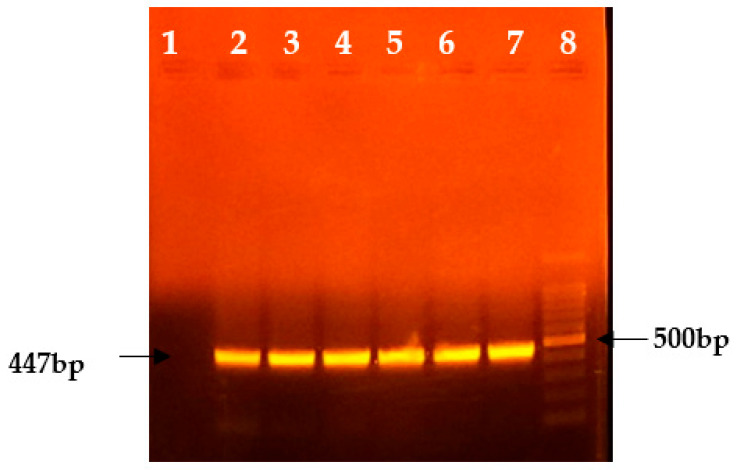
PCR gel picture of the fusion gene (447 bp). Lane 1: Negative control; Lanes 2, 3, 4, 5, 6, and 7: Positive samples; Lane 8: DNA ladder (100 bp).

**Table 1 viruses-17-00293-t001:** Molecular screening for the P32 Envelop gene of samples with 95% confidence interval.

S. No	Sample Type	Total	Positive	Negative
1.	Blood	116	00 (0.0%, 0.0–3.13%)	116 (100%, 96.9–100%)
2.	Swab	47	03 (6.4%, 1.3–17.5%)	44 (93.6%, 82.5–98.7%)
3.	Tissue	26	18 (69.2%, 48.2–85.7%)	08 (30.8%, 14.3–51.8%)

**Table 2 viruses-17-00293-t002:** Molecular screening for the fusion gene with 95% confidence interval.

S. No	Sample Type	Total	Positive	Negative
1.	Blood	00	00	00
2.	Swab	03	02 (66.7%, 9.4–99.2%)	01 (33.3%, 0.8–90.6%)
3.	Tissue	18	14 (77.8%, 52.4–93.6%)	04 (22.2%, 6.4–47.6%)

**Table 3 viruses-17-00293-t003:** The association between seroprevalence of LSD in cattle and animal characteristics studied during the present study.

Parameters		Sample	Seropositive	Seronegative	χ^2^ Value	*p*-Value
Breed	HFC	22	7	15	9.552	*p* > 0.05
	JC	58	15	43		
	ND	62	12	50		
	OC	8	0	8		
	Sahiwal	17	1	16		
	Gir	17	1	16		
	Total	184	36	148		
Age	<2 years	52	10	42	3.01	*p* > 0.05
	2–4 years	57	10	47		
	>4 years	75	16	59		
	Total	184	36	148		
Sex	Male	49	10	39	8.38	*p* < 0.05
	Female	135	26	109		
	Total	184	36	148		
Immunization statusS	Immunized	37	0	37	9.76	*p* < 0.05
	Non-immunized	147	36	111		
	Total	184	36	148		

**Table 4 viruses-17-00293-t004:** Results of biochemical analysis for PCR-positive samples.

S. No.	Sample	Creatinine (mg/dL)	SGPT (U/L)	SGOT (U/L)	BUN (mg/dL)	Total Protien (g/dL)	Albu. (g/dL)	Globu. (g/dL)	A: G (g/dL)
1	S-02	0.96	90.17	99.01	29.63	6.87	5.36	1.51	3.550
2	S-04	0.93	33.59	49.5	21.11	7.45	3.57	3.88	0.920
3	S-09	1.26	76.02	83.1	15.69	7.81	4.83	2.98	1.621
4	S-42	1.37	160.9	176.8	17.45	5.17	4.36	0.81	5.383
5	NS-01	2.84	N/A	N/A	21.92	8.19	4.82	3.37	1.430
6	NS-02	1.55	270.5	272.3	13.26	6.76	4.76	2	2.380
Mean of Positive Samples		1.485	126.236	136.142	19.843	7.042	4.617	2.425	2.547
Normal Values of Healthy Cattles		0.7–1.1	11.0–40.0	78–132	10.0–26.0	5.9–7.7	2.7–4.3	2.5–4.1	0.6–1.6

N/A Not available.

**Table 5 viruses-17-00293-t005:** A 95% confidence interval along with the mean ± SD of the biochemical profiles in LSDV-positive samples and in LSDV-negative samples.

			Result (Mean ± SD)		
Parameters	Positive Samples	95% CI Values	Negative Samples	95% CI Values	*p*-Value
Creatinine	1.48 ± 0.705	74–223%	1.13 ± 0.307	80–145%	0.4476
SGPT	126.23 ± 92.8	288–22,358%	49.5 ± 20.8	2767–7132%	0.2197
SGOT	136.14 ± 89.3	4243–22,984%	85.45 ± 37.5	4609–12,480%	0.3967
BUN	26.6 ± 16.0	1376–2591%	19.8 ± 5.80	1371–2588%	0.367
Total protein	7.04 ± 1.07	592–816%	5.39 ± 2.11	317–760%	0.1489
Albumin	4.62 ± 0.604	3.98–5.25%	3.98 ± 0.642	330–465%	0.1864
Globulin	2.42 ± 1.18	118–366%	1.41 ± 1.65	32–314%	0.3155
A: G	6.32 ± 4.63	80–429%	2.55 ± 1.66	80–429%	0.1025

## Data Availability

Data available in a publicly accessible repository. The data presented in the study are openly available in NCBI GenBank database under the accession numbers OQ183330.1, OQ150023.1, OR228441.1, OR228442.1, OR228443.1, OR228444.1, and OR228445.1.

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
