# Peer review of "Molecular Detection, Seroprevalence and Biochemical Analysis of Lumpy Skin Disease Virus"

_viruses, 2025, doi:10.3390/v17030293_

Round 1
Reviewer 1 Report
Comments and Suggestions for Authors
This study reports the LSDV detection, seroprevalence and biochemical analysis in 6 serum samples of cattle from India. The study included 189 samples in total, analysed by conventional PCR and sequenced. However, no phylogenetic analysis was made. Serum samples, in total 184 , were analysed with ELISA.
Molecular epidemiological investigations are appreciated, they provide valuable information about LSDV strains distribution in different geographical areas. In this context, the detection and distribution of LSDV strains in India would be valuable. Unfortunately the authors did not provide these details.
Unfortunately, this study is not suitable for publication in the present form. Significant improvement is needed, precisely in the following:
- Please provide a clear aim for your study – the current version lacks focus, it starts with molecular investigation but ends with discussing biochemical changes in serum. Significant improvement would be made if you would just focus on virus prevalence, showing which strains you detected and placing them in phylogenetical context with other strains. Your findings on the biochemical changes can not be related exclusively to LSD, to make this kind of analysis completely different design of the study is needed.
-significant gaps are identified in Material and Method section -A number of details about sample collection is missing, species, age, breed, health status. Was the sampling performed in the frame of a program, suspicion or something else. How did you select sample size? Please specify which tissues were collected, why and how? Please indicate how many animals were included in the study.
Other suggestions:
Line 38 : LSD is caused with different strains of LSDV, Neethling is one of the LSDV strains. Please check the available reports on the circulation of different strains in Asia and consider placing your detected strains in this context.
Table 3 Please refer to seroprevalence as this table is showing seroprevalence data
Discussion
Lines 210-216 – One of the LSDV characteristics is that it can be intermittently detected in the blood, which could explain your negative finding. Moreover, at low levels LSDV can be found in the blood and this can be nicely detected with real-time PCR which offers higher sensitivity than classical PCR. This is also indicated in WOAH chapter on LSDV.
Line 218-229 Different breeds were included in this study. However, the number of representatives for each breed is variable and for certain breeds very low. In my opinion, this pile of data does not support drawing any conclusions about the prevalence of LSDV among different breeds.
I listed only a number of important areas for improvement. By addressing these areas for improvement, the study would become more robust, informative, and suitable for publication in a scientific journal.
Reviewer 2 Report
Comments and Suggestions for Authors
Line 38: LSD is cause by a Neethling strain LSD virus
Comment: LSD is not only caused by the Neethling strain. This is only the prototype LSDV strain and has been studied the most.
2.1
If possible, mention in this section some history of LSD cases in the area sampled (this is relevant as the study shows a surprising high level of PCR positivity which wouldn’t be expected if animals had not been recently infected; how many outbreaks had been reported, how recent were they, was sampling done from animals having any clinical signs etc. ?
Line 93
Concerning the primer sequence, reference is made to the OIE Manual, but number 11 under Reference doesn’t refer to such. Please correct.
Tables 1 & 2
Suggestion: for column 2, rather use Sample type, column 3: Total; column 4 : Positive, column 5: Negative.
Also, for all tables showing count data, it would be advisable to include the percentage values in the table as well as the 95% confidence intervals (for proportions, t distribution). Mention inclusion of percentage and 95% CI values in the table heading. Can use Epitools to assist (epitools.ausvet.com.au).
e.g. for Table 1
|
Sample type |
Total |
Positive |
Negative |
|
|
|
|
|
|
Blood |
116 |
0 (1.7%; 0.0%-3.13%)
|
116 (100%; 96.9%-100%) |
|
Nasal swabs |
47 |
3 (6.4%; 1.3%-17.5%) |
44 (93.6%; 82.5%-98.7%) |
|
Tissue |
26 |
18 (69.2%; 48.2%-85.7%) |
8 (30.8%; 14.3%-51.8%) |
In this case, the Clopper-Pearson exact method was used for CI determinations. Note conventional formula not used when dealing with 0% or 100% values.
Tabler 3. Mention in the table legend that the data refers to serological findings. Also, it would be an association between seropositivity/seronegativity (rather than LSD) and various animal characteristics
Table 4
Also include the 95% confidence intervals for the mean values. This will also assist when comparing with the normal range (which should be the 95% interval for a normal distribution). The legend for Table 4 needs to indicate whether these findings were for seropositive or PCR positive samples. Also clarify what the colour code refers to.
Table 5: Suggestion: 95% CI is more informative than SDs
Foer A:G values in Table 5, there is a discrepancy with regards to the positive samples and what is indicted in Table 4. Please correct Also, for the columns, indicate PCR positive & PCR negative samples, or was it seropositive & seronegative samples ?
I am not sure whether all the biochemical blood data included is necessary, as it isn’t particularly informative in this study. Perhaps reduce, cut out, combine, summarise graphically, ... (?)
Line 196:
Comment : since the overall positivity rate is given for the P32 assays, why not also for the fusion gene PCR ?
Line 198: .[14], with a rate of 87.5% for scab samples; [15] reporting 80% for scab samples;......
Comment: This sentence is confusing. Presumably a comparison is being made between different published findings in the literature. To make it clearer to follow, perhaps mention the author names as well, with their findings.
Line 202 to 205. Determine which of the findings were based on real-time PCR assays and which weren’t. One can expect hydrolysis probe-based real-time assays to be substantially more sensitive than conventional assays. Perhaps a log or more.
Line 210. Include the 95% confidence interval for this finding.
Line 238
Comment: Higher seroprevalence in adult cattle could also be due to a longer time period for possible exposure than younger animals, assuming outbreaks had already occurred several years previously
Line 243
seropositivity to Lumpy skin disease..
Suggestion:. seropositivity to lumpy skin disease virus
Line 246: 95% confidence intervals take sample sizes into account, hence also include when reporting and comparing differences
Line 254: ... the biochemical profile provides valuable insights into the pathogenesis and prognosis
Comment: is this really the case ? This study doesn’t seem to indicate statistically significant findings
Line 250 to 304: devoted to discussing the biochemical blood profiles.
These seems excessive considering that the findings were based on small sample numbers and results were largely statistically insignificant, Could the discussion be reduced to one or two paragraphs ?
References
Check details under references
Line 360: ...Lumpy skin Disease Virus....
What journal ?
15. Plos one
Change to Plos One
25: Acta tropica
Change to Acta Tropica
32: Veterinary world
Change to : Veterinary World
37 and 41: italicise journals names as for the rest
Comments on the Quality of English Language
Title
The title needs adjustment. A biochemical analysis is not done on a virus
A suggestion, perhaps : Diagnostic test findings in Indian Cattle following Infection with Lumpy Skin Disease Virus
Capitals: don’t capitalise disease or virus names, except if a proper noun
Line 10: Lumpy Skin Disease (line 10, 58, 69), Lumpy skin disease virus (line 69 and elsewhere), Goatpox virus (line 46), Sheeppox virus (line 46)
and elsewhere (eg Line 60 to 61).
Exceptions are when classifying and naming viral species (rather than referring to the physical virus itself), thus for classification Lumpy skin disease virus. Can refer to virus nomenclature guidelines by the ICTV.
Thus: lumpy skin disease, lumpy skin disease virus, goatpox virus, sheeppox virus,
bovine herpesvirus
Line 10 &11
... caused by the capripoxvirus genus of the Poxviridae family
Comment: LSD is not caused by a genus. Italicise genus and family names
Suggestion: caused by lumpy skin disease virus (LSDV), belonging to the Capripoxvirus genus and Poxviridae family.
Line 12
...biochemical analysis of LSD.
Comment: biochemical analysis of the disease is not done but on samples from an animal
Suggestion: This study reports on the molecular detection and seroprevalence of LSDV and biochemical analysis of samples from previously infected cattle....
Line 22: ...for seroprevalence study
Suggestion... for seroprevalence studies.
Line 27: ... and possess a serious threat to livestock industry resulting in significant financial losses
Suggestion: ... and possesses a serious threat to the livestock industry resulting in a significant financial loss...
Line 36: ..as Southern states..
Suggestion: ... southern states..
Linbe 69: ... Cattle..
Suggestion:. Cattle
Line 89: ... presence of P32 gene and fusion gene of LSDV.
Suggestion: ... presence of the P32 and fusion genes of LSDV.
Line 90: ...targeting P32 gene,..
Suggestion: .. targeting the P32 gene
Line 76 : Total 189 samples
Suggestion: A total of 189 samples...
Line 92: .. .that encodes the amplicon of 192 bp..
Suggestion: that yields a 192 bp amplicon
Line 93:... was performed in 25ul volume
Suggestion: . was performed using a 25ul reaction volume
Line 96: .. performed in Bio-Rad Thermo cycler
Suggestion: ...was performed in a Bio-Rad thermocycler
Line 116: ...the Optical Density value...
Suggestion... the optical density ...
Line 120: Analysis of Serum Biochemical Parameters
Suggestion: Analysis of serum biochemical parameters
Line 121: Creatinine, serum glutamic Pyruvic Transaminase (SGPT) ...... Serum glutamic oxaloacetic Transaminase test SGPT).....
Suggestion: Creatinine, serum glutamic pyruvic transaminase (SGOT) ...... serum glutamic oxaloacetic transaminase test (SGPT).....
Line 135: ...subjected for conventional PCR targeting p32 gene the expected product size was 192 bp (figure 01).... (figure 02).
Suggestion: ..subjected to conventional PCR targeting the P32 gene, with an expected product size of 192 bp (Figure 01).... (Figure 02).
Line 151: ... were submitted in
Suggestion: ... were submitted to the ....
Line 155: ..seroprevalence of LSDV was found to be
Suggestion: ...seroprevalence in cattle for LSDV
Line 185: The serum biochemical values for Lumpy skin disease virus (LSDV)-positive samples..
Suggestion: The serum biochemical values for samples from LSDV seropositve (or was it PCR positive ?) cattle.
Line 194: of the p32 gene
Suggestion: ..P 32 gene
Round 2
Reviewer 2 Report
Comments and Suggestions for Authors
Line 198: Table 5. 95% Confidence Interval along with Mean ± SD of the biochemical profiles in LSDV positive samples and in negative samples.
Suggestion: Table 5. 95% Mean ± SD and 95% confidence intervals for the biochemical test results for LSDV positive and negative samples.
Comment: Reformat the table. Remove the top row. Also the include units for the parameters and add ‘PCR’ for the headings, ie PCR positive samples, PCR negative samples.
Tabe 3
Include percentage values and 95% CI’s in Table 3 (for the Seropositive and Seronegative columns) as was done for Tables 1 & 2.
Table 5. The CI’s shown in this case should not be percentages (they would have the same units as the means). Thus, for creatine the 95% CI is 0.745 - 2.225. Under section 2.6, mention the number of negative samples that were tested.
Is there merit still having Table 4, apart from the normal ranges ?
Alternatively, if the number of negative samples are not too large, the individual negative values could be incorporated into Table 4, perhaps below the positives.
If so, then remove Table 5 and in Table 4 for the row ‘Mean of positive samples’, replace with ‘Mean of PCR positive samples± SD and 95% CI’s ',
as shown below for the Creatinine column,
|
Mean (or use the symbol xÌ„) of PCR positive samples ±SD, (95% CI) |
1.485±0.705 (0.745, 2.225) |
|
and do similarly for the negative samples.
Comments on the Quality of English Language
Tite: Diagnostic test findings in Indian Cattle following infection with lumpy skin disease virus
Comment: Use capitals accordingly as for titles of articles i.e. for major words, except articles, prepositions and conjunctions. Thus:
Diagnostic Test Findings in Indian Cattle Following Infection with Lumpy Skin Disease Virus
Line 37: also affected North-Western states like Gujarat, Rajasthan…
Suggestion: also affected north-western states like Gujarat, Rajasthan..
Line 144: Table 1. Molecular screening for P32 Envelop gene of samples with 95% Confidence Interval as below:
Suggestion: Table 1. PCR test result numbers for the P32 envelope gene, with percentages and 95% confidence intervals included.
Line 149: Table 2. Molecular Screening for Fusion gene with 95% Confidence Interval.
Suggestion: PCR test result numbers for the fusion gene, with percentages and 95% confidence intervals included.
Line 152: Figure 1. PCR gel picture of P32 gene (192bp).
Suggestion: Figure 1. Electrophoretic analysis following P32 gene PCR (192 bp amplicon).
Line 155: Figure 2. PCR gel picture of fusion gene (447bp)
Suggestion: Figure 2. Electrophoretic analysis following fusion gene PCR (447bp amplicon)
Line 170: Table 3. Association between seroprevalence of LSD in cattle and animal characteristics studied during present study.
Suggestion: Association between seroprevalence of LSD in cattle and selected animal characteristics.
Line 193: Table 04. Result of Biochemical analysis for PCR positive samples
Suggestion: Table 04. Test results following biochemical analysis of PCR positive samples
